# Variational Consensus Monte Carlo

**Maxim Rabinovich, Elaine Angelino, and Michael I. Jordan**
Computer Science Division
University of California, Berkeley
`{rabinovich, elaine, jordan}@eecs.berkeley.edu`

## Abstract

Practitioners of Bayesian statistics have long depended on Markov chain Monte Carlo (MCMC) to obtain samples from intractable posterior distributions. Unfortunately, MCMC algorithms are typically serial, and do not scale to the large datasets typical of modern machine learning. The recently proposed consensus Monte Carlo algorithm removes this limitation by partitioning the data and drawing samples conditional on each partition in parallel [22]. A fixed aggregation function then combines these samples, yielding approximate posterior samples. We introduce *variational consensus Monte Carlo* (VCMC), a variational Bayes algorithm that optimizes over aggregation functions to obtain samples from a distribution that better approximates the target. The resulting objective contains an intractable entropy term; we therefore derive a relaxation of the objective and show that the relaxed problem is blockwise concave under mild conditions. We illustrate the advantages of our algorithm on three inference tasks from the literature, demonstrating both the superior quality of the posterior approximation and the moderate overhead of the optimization step. Our algorithm achieves a relative error reduction (measured against serial MCMC) of up to 39% compared to consensus Monte Carlo on the task of estimating 300-dimensional probit regression parameter expectations; similarly, it achieves an error reduction of 92% on the task of estimating cluster comembership probabilities in a Gaussian mixture model with 8 components in 8 dimensions. Furthermore, these gains come at moderate cost compared to the runtime of serial MCMC—achieving near-ideal speedup in some instances.

## 1 Introduction

Modern statistical inference demands scalability to massive datasets and high-dimensional models. Innovation in distributed and stochastic optimization has enabled parameter estimation in this setting, e.g. via stochastic [3] and asynchronous [20] variants of gradient descent. Achieving similar success in Bayesian inference – where the target is a posterior distribution over parameter values, rather than a point estimate – remains computationally challenging.

Two dominant approaches to Bayesian computation are variational Bayes and Markov chain Monte Carlo (MCMC). Within the former, scalable algorithms like stochastic variational inference [11] and streaming variational Bayes [4] have successfully imported ideas from optimization. Within MCMC, adaptive subsampling procedures [2, 14], stochastic gradient Langevin dynamics [25], and Firefly Monte Carlo [16] have applied similar ideas, achieving computational gains by operating only on data subsets. These algorithms are serial, however, and thus cannot take advantage of multicore and multi-machine architectures. This motivates data-parallel MCMC algorithms such as asynchronous variants of Gibbs sampling [1, 8, 12].

Our work belongs to a class of *communication-avoiding* data-parallel MCMC algorithms. These algorithms partition the full dataset $X_{1:N}$ into $K$ disjoint subsets $X_{I_{1:K}}$ where $X_{I_k}$ denotes the data

associated with core $k$. Each core samples from a *subposterior* distribution,

$$p_k(\theta_k) \propto p(X_{I_k} \mid \theta_k) \, p(\theta_k)^{1/K}, \tag{1}$$

and then a centralized procedure combines the samples into an approximation of the full posterior. Due to their efficiency, such procedures have recently received substantial attention [18, 22, 24].

One of these algorithms, *consensus Monte Carlo* (CMC), requires communication only at the start and end of sampling [22]. CMC proceeds from the intuition that subposterior samples, when aggregated correctly, can approximate full posterior samples. This is formally backed by the factorization

$$p(\theta \mid x_{1:N}) \propto p(\theta) \prod_{k=1}^{K} p(X_{I_k} \mid \theta) = \prod_{k=1}^{K} p_k(\theta). \tag{2}$$

If one can approximate the subposterior densities $p_k$, using kernel density estimates for instance [18], it is therefore possible to recombine them into an estimate of the full posterior.

Unfortunately, the factorization does not make it immediately clear how to aggregate on the level of *samples* without first having to obtain an estimate of the densities $p_k$ themselves. CMC alters (2) to untie the parameters across partitions and plug in a deterministic link $F$ from the $\theta_k$ to $\theta$:

$$p(\theta \mid x_{1:N}) \approx \prod_{k=1}^{K} p_k(\theta_k) \cdot \delta_{\theta=F(\theta_1,\ldots,\theta_K)}. \tag{3}$$

This approximation and an aggregation function motivated by a Gaussian approximation lie at the core of the CMC algorithm [22].

The introduction of CMC raises numerous interesting questions whose answers are essential to its wider application. Two among these stand out as particularly vital. First, how should the aggregation function be chosen to achieve the closest possible approximation to the target posterior? Second, when model parameters exhibit structure or must conform to constraints — if they are, for example, positive semidefinite covariance matrices or labeled centers of clusters — how can the weighted averaging strategy of Scott et al. [22] be modified to account for this structure?

In this paper, we propose *variational consensus Monte Carlo* (VCMC), a novel class of data-parallel MCMC algorithms that allow both questions to be addressed. By formulating the choice of aggregation function as a variational Bayes problem, VCMC makes it possible to adaptively choose the aggregation function to achieve a closer approximation to the true posterior. The flexibility of VCMC likewise supports nonlinear aggregation functions, including structured aggregation functions applicable to not purely vectorial inference problems.

An appealing benefit of the VCMC point of view is a clarification of the untying step leading to (3). In VCMC, the approximate factorization corresponds to a variational approximation to the true posterior. This approximation can be viewed as the joint distribution of $(\theta_1,\ldots,\theta_K)$ and $\theta$ in an augmented model that assumes conditional independence between the data partitions and posits a deterministic mapping from partition-level parameters to the single global parameter. The added flexibility of this point-of-view makes it possible to move beyond subposteriors and include alternative forms of (3) within the CMC framework. In particular, it is possible to define $p_k(\theta_k) = p(\theta_k) \, p(X_{I_k} \mid \theta_k)$, using partial posteriors in place of subposteriors (cf. [23]). Although extensive investigation of this issue is beyond the scope of this paper, we provide some evidence in Section 6 that partial posteriors are a better choice in some circumstances and demonstrate that VCMC can provide substantial gains in both the partial posterior and subposterior settings.

Before proceeding, we outline the remainder of this paper. Below, in §2, we review CMC and related data-parallel MCMC algorithms. Next, we cast CMC as a variational Bayes problem in §3. We define the variational optimization objective in §4, addressing the challenging entropy term by relaxing it to a concave lower bound, and give conditions for which this leads to a blockwise concave maximization problem. In §5, we define several aggregation functions, including novel ones that enable aggregation of structured samples—e.g. positive semidefinite matrices and mixture model parameters. In §6, we evaluate the performance of VCMC and CMC relative to serial MCMC. We replicate experiments carried out by Scott et al. [22] and execute more challenging experiments in higher dimensions and with more data. Finally in §7, we summarize our approach and discuss several open problems generated by this work.

## 2 Related work

We focus on data-parallel MCMC algorithms for large-scale Bayesian posterior sampling. Several recent research threads propose schemes in the setting where the posterior factors as in (2). In general, these parallel strategies are approximate relative to serial procedures, and the specific algorithms differ in terms of the approximations employed and amount of communication required.

At one end of the communication spectrum are algorithms that fit into the MapReduce model [7]. First, $K$ parallel cores sample from $K$ subposteriors, defined in (1), via any Monte Carlo sampling procedure. The subposterior samples are then aggregated to obtain approximate samples from the full posterior. This leads to the challenge of designing proper and efficient aggregation procedures.

Scott et al. [22] propose *consensus Monte Carlo* (CMC), which constructs approximate posterior samples via weighted averages of subposterior samples; our algorithms are motivated by this work. Let $\theta_{k,t}$ denote the $t$-th subposterior sample from core $k$. In CMC, the aggregation function averages across each set of $K$ samples $\{\theta_{k,t}\}_{k=1}^{K}$ to produce one approximate posterior sample $\hat{\theta}_t$. Uniform averaging is a natural but naïve heuristic that can in fact be improved upon via a weighted average,

$$\hat{\theta} = F\left(\theta_{1:K}\right) = \sum_{k=1}^{K} W_k \theta_k, \tag{4}$$

where in general, $\theta_k$ is a vector and $W_k$ can be a matrix. The authors derive weights motivated by the special case of a Gaussian posterior, where each subposterior is consequently also Gaussian. Let $\Sigma_k$ be the covariance of the $k$-th subposterior. This suggests weights $W_k = \Sigma_k^{-1}$ equal to the subposteriors' inverse covariances. CMC treats arbitrary subpostertiors as Gaussians, aggregating with weights given by empirical estimates of $\hat{\Sigma}_k^{-1}$ computed from the observed subposterior samples.

Neiswanger et al. [18] propose aggregation at the level of distributions rather than samples. Here, the idea is to form an approximate posterior via a product of density estimates fit to each subposterior, and then sample from this approximate posterior. The accuracy and computational requirements of this approach depend on the complexity of these density estimates. Wang and Dunson [24] develop alternate data-parallel MCMC methods based on applying a Weierstrass transform to each subposterior. These *Weierstrass sampling* procedures introduce auxiliary variables and additional communication between computational cores.

## 3 Consensus Monte Carlo as variational inference

Given the distributional form of the CMC framework (3), we would like to choose $F$ so that the induced distribution on $\theta$ is as close as possible to the true posterior. This is precisely the problem addressed by variational Bayes, which approximates an intractable posterior $p\left(\theta \mid \mathrm{X}\right)$ by the solution $q^*$ to the constrained optimization problem

$$\min D_{\mathrm{KL}}\left(q \,\|\, p\left(\cdot \mid \mathrm{X}\right)\right) \quad \text{subject to} \ \ q \in \mathcal{Q},$$

where $\mathcal{Q}$ is the family of variational approximations to the distribution, usually chosen to make both optimization and evaluation of target expectations tractable. We thus view the aggregation problem in CMC as a variational inference problem, with the variational family given by all distributions $\mathcal{Q} = \mathcal{Q}_{\mathcal{F}} = \{q_F \colon F \in \mathcal{F}\}$, where each $F$ is in some function class $\mathcal{F}$ and defines a density

$$q_F\left(\theta\right) = \int_{\Omega^K} \prod_{k=1}^{K} p_k\left(\theta_k\right) \cdot \delta_{\theta=F(\theta_1,\dots,\theta_K)} \, \mathrm{d}\theta_{1:K}.$$

In practice, we optimize over finite-dimensional $\mathcal{F}$ using projected stochastic gradient descent (SGD).

## 4 The variational optimization problem

Standard optimization of the variational Bayes objective uses the evidence lower bound (ELBO)

$$\log p\left(\mathrm{X}\right) = \log \mathbb{E}_q\left[\frac{p\left(\theta, \mathrm{X}\right)}{q\left(\theta\right)}\right] \geq \mathbb{E}_q\left[\log \frac{p\left(\theta, \mathrm{X}\right)}{q\left(\theta\right)}\right]$$

$$= \log p\left(\mathrm{X}\right) - D_{\mathrm{KL}}\left(q \,\|\, p\left(\cdot \mid \mathrm{X}\right)\right) =: \mathcal{L}_{\mathrm{VB}}\left(q\right). \tag{5}$$

We can therefore recast the variational optimization problem in an equivalent form as

$$\max \mathcal{L}_{\mathrm{VB}}(q) \quad \text{subject to} \quad q \in \mathcal{Q}.$$

Unfortunately, the variational Bayes objective $\mathcal{L}_{\mathrm{VB}}$ remains difficult to optimize. Indeed, by writing

$$\mathcal{L}_{\mathrm{VB}}(q) = \mathbb{E}_q \left[\log p(\theta, \mathrm{X})\right] + \mathrm{H}[q]$$

we see that optimizing $\mathcal{L}_{\mathrm{VB}}$ requires computing an entropy $\mathrm{H}[q]$ and its gradients. We can deal with this issue by deriving a lower bound on the entropy that relaxes the objective further.

Concretely, suppose that every $F \in \mathcal{F}$ can be decomposed as $F(\theta_{1:K}) = \sum_{k=1}^{K} F_k(\theta_k)$, with each $F_k$ a differentiable bijection. Since the $\theta_k$ come from subposteriors conditioning on different segments of the data, they are independent. The entropy power inequality [6] therefore implies

$$\mathrm{H}[q] \geq \max_{1 \leq k \leq K} \mathrm{H}[F_k(\theta_k)] = \max_{1 \leq k \leq K} \left(\mathrm{H}[p_k] + \mathbb{E}_{p_k}\left[\log \det\left[J(F_k)(\theta_k)\right]\right]\right)$$

$$\geq \min_{1 \leq k \leq K} \mathrm{H}[p_k] + \max_{1 \leq k \leq K} \mathbb{E}_{p_k}\left[\log \det\left[J(F_k)(\theta_k)\right]\right] \tag{6}$$

$$\geq \min_{1 \leq k \leq K} \mathrm{H}[p_k] + \frac{1}{K}\sum_{k=1}^{K} \mathbb{E}_{p_k}\left[\log \det\left[J(F_k)(\theta_k)\right]\right] =: \tilde{\mathrm{H}}[q], \tag{7}$$

where $J(f)$ denotes the Jacobian of the function $f$. The proof can be found in the supplement.

This approach gives an explicit, easily computed approximation to the entropy—and this approximation is a lower bound, allowing us to interpret it simply as a further relaxation of the original inference problem. Furthermore, and crucially, it decouples $p_k$ and $F_k$, thereby making it possible to optimize over $F_k$ without estimating the entropy of any $p_k$. We note additionally that if we are willing to sacrifice concavity, we can use the tighter lower bound on the entropy given by (6).

Putting everything together, we can define our relaxed variational objective as

$$\mathcal{L}(q) = \mathbb{E}_q\left[\log p(\theta, \mathrm{X})\right] + \tilde{\mathrm{H}}[q]. \tag{8}$$

Maximizing this function is the variational Bayes problem we consider in the remainder of the paper.

**Conditions for concavity**  Under certain conditions, the problem posed above is blockwise concave. To see when this holds, we use the language of graphical models and exponential families. To derive the result in the greatest possible generality, we decompose the variational objective as

$$\mathcal{L}_{\mathrm{VB}} = \mathbb{E}_q\left[\log p(\theta, \mathrm{X})\right] + \mathrm{H}[q] \geq \tilde{\mathcal{L}} + \tilde{\mathrm{H}}[q]$$

and prove concavity directly for $\tilde{\mathcal{L}}$, then treat our choice of relaxed entropy (7). We emphasize that while the entropy relaxation is only defined for decomposed aggregation functions, concavity of the partial objective holds for arbitrary aggregation functions. All proofs are in the supplement.

Suppose the model distribution is specified via a graphical model $G$, so that $\theta = (\theta^u)_{u \in V(G)}$, such that each conditional distribution is defined by an exponential family

$$\log p\left(\theta^u \mid \theta^{\mathrm{par}(u)}\right) = \log h^u(\theta^u) + \sum_{u' \in \mathrm{par}(u)} \left(\theta^{u'}\right)^T T^{u' \to u}(\theta^u) - \log A^u\left(\theta^{\mathrm{par}(u)}\right).$$

If each of these log conditional density functions is log-concave in $\theta^u$, we can guarantee that the log likelihood is concave in each $\theta^u$ individually.

**Theorem 4.1** (Blockwise concavity of the variational cross-entropy). *Suppose that the model distribution is specified by a graphical model $G$ in which each conditional probability density is a log-concave exponential family. Suppose further that the variational aggregation function family satisfies $\mathcal{F} = \prod_{u \in V(G)} \mathcal{F}^u$ such that we can decompose each aggregation function across nodes via*

$$F(\theta) = (F^u(\theta^u))_{u \in V(G)}, \quad F \in \mathcal{F} \quad \text{and} \quad F^u \in \mathcal{F}^u.$$

*If each $\mathcal{F}^u$ is a convex subset of some vector space $\mathcal{H}^u$, then the variational cross-entropy $\tilde{\mathcal{L}}$ is concave in each $F^u$ individually.*

Assuming that the aggregation function can be decomposed into a sum over functions of individual subposterior terms we can also prove concavity of our entropy relaxation (7).

**Theorem 4.2** (Concavity of the relaxed entropy). *Suppose $\mathcal{F} = \prod_{k=1}^{K} \mathcal{F}_k$, with each function $F \in \mathcal{F}$ decomposing as $F(\theta_1, \ldots, \theta_K) = \sum_{k=1}^{K} F_k(\theta_k)$ for unique bijective $F_k \in \mathcal{F}_k$. Then the relaxed entropy (7) is concave in $F$.*

As a result, we derive concavity of the variational objective in a broad range of settings.

**Corollary 4.1** (Concavity of the variational objective). *Under the hypotheses of Theorems 4.1 and 4.2, the variational Bayes objective $\mathcal{L} = \tilde{\mathcal{L}} + \tilde{H}$ is concave in each $F^u$ individually.*

## 5 Variational aggregation function families

The performance of our algorithm depends critically on the choice of aggregation function family $\mathcal{F}$. The family must be sufficiently simple to support efficient optimization, expressive to capture the complex transformation from the set of subposteriors to the full posterior, and structured to preserve structure in the parameters. We now illustrate some aggregation functions that meet these criteria.

**Vector aggregation.** In the simplest case, $\theta \in \mathbb{R}^d$ is an unconstrained vector. Then, a linear aggregation function $F_W = \sum_{k=1}^{K} W_k \theta_k$ makes sense, and it is natural to impose constraints to make this sum behave like a weighted average—i.e., each $W_k \in \mathcal{S}_+^d$ is a positive semidefinite (PSD) matrix and $\sum_{k=1}^{K} W_k = I_d$. For computational reasons, it is often desirable to restrict to diagonal $W_k$.

**Spectral aggregation.** Cases involving structure exhibit more interesting behavior. Indeed, if our parameter is a PSD matrix $\Lambda \in \mathcal{S}_+^d$, applying the vector aggregation function above to the flattened vector form $\mathrm{vec}(\Lambda)$ of the parameter does not suffice. Denoting elementwise matrix product as $\circ$, we note that this strategy would in general lead to $F_W(\Lambda_{1:m}) = \sum_{k=1}^{K} W_k \circ \Lambda_k \notin \mathcal{S}_+^d$.

We therefore introduce a more sophisticated aggregation function that preserves PSD structure. For this, given symmetric $A \in \mathbb{R}^{d \times d}$, define $R(A)$ and $D(A)$ to be orthogonal and diagonal matrices, respectively, such that $A = R(A)^T D(A) R(A)$. Impose further—and crucially—the canonical ordering $D(A)_{11} \geq \cdots \geq D(A)_{dd}$. We can then define our spectral aggregation function by

$$F_W^{\mathrm{spec}}(\Lambda_{1:K}) = \sum_{k=1}^{K} R(\Lambda_k)^T [W_k D(\Lambda_k)] R(\Lambda_k).$$

Assuming $W_k \in \mathcal{S}_+^d$, the output of this function is guaranteed to be PSD, as required. As above we restrict the set of $W_k$ to the matrix simplex $\{(W_k)_{k=1}^{K} : W_k \in \mathcal{S}_+^d, \ \sum_{k=1}^{K} W_k = I\}$.

**Combinatorial aggregation.** Additional complexity arises with unidentifiable latent variables and, more generally, models with multimodal posteriors. Since this class encompasses many popular algorithms in machine learning, including factor analysis, mixtures of Gaussians and multinomials, and latent Dirichlet allocation (LDA), we now show how our framework can accommodate them.

For concreteness, suppose now that our model parameters are given by $\theta \in \mathbb{R}^{L \times d}$, where $L$ denotes the number of global latent variables (e.g. cluster centers). We introduce discrete alignment parameters $a_k$ that indicate how latent variables associated with partitions map to global latent variables. Each $a_k$ is thus a one-to-one correspondence $[L] \to [L]$, with $a_{k\ell}$ denoting the index on worker core $k$ of cluster center $\ell$. For fixed $a$, we then obtain the variational aggregation function

$$F_a(\theta_{1:K}) = \left( \sum_{k=1}^{K} W_{k\ell} \theta_{k a_{k\ell}(\ell)} \right)_{\ell=1}^{L}.$$

Optimization can then proceed in an alternating manner, switching between the alignments $a_k$ and the weights $W_k$, or in a greedy manner, fixing the alignments at the start and optimizing the weight matrices. In practice, we do the latter, aligning using a simple heuristic objective $\mathcal{O}(a) = \sum_{k=2}^{K} \sum_{\ell=1}^{L} \left\| \bar{\theta}_{k a_{k\ell}} - \bar{\theta}_{1\ell} \right\|_2^2$, where $\bar{\theta}_{k\ell}$ denotes the mean value of cluster center $\ell$ on partition $k$. As $\mathcal{O}$ suggests, we set $a_{1\ell} = \ell$. Minimizing $\mathcal{O}$ via the Hungarian algorithm [15] leads to good alignments.

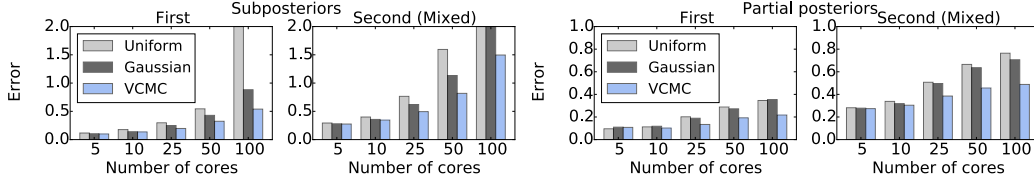

Figure 1: High-dimensional probit regression ($d = 300$). Moment approximation error for the uniform and Gaussian averaging baselines and VCMC, relative to serial MCMC, for subposteriors *(left)* and partial posteriors *(right)*; note the different vertical axis scales. We assessed three groups of functions: first moments, with $f(\beta) = \beta_j$ for $1 \leq j \leq d$; pure second moments, with $f(\beta) = \beta_j^2$ for $1 \leq j \leq d$; and mixed second moments, with $f(\beta) = \beta_i\beta_j$ for $1 \leq i < j \leq d$. For brevity, results for pure second moments are relegated to Figure 5 in the supplement.

## 6 Empirical evaluation

We now evaluate VCMC on three inference problems, in a range of data and dimensionality conditions. In the vector parameter case, we compare directly to the simple weighting baselines corresponding to previous work on CMC [22]; in the other cases, we compare to structured analogues of these weighting schemes. Our experiments demonstrate the advantages of VCMC across the whole range of model dimensionality, data quantity, and availability of parallel resources.

**Baseline weight settings.** Scott et al. [22] studied linear aggregation functions with fixed weights,

$$W_k^{\mathrm{unif}} = \frac{1}{K} \cdot I_d \qquad \text{and} \qquad W_k^{\mathrm{gauss}} \propto \mathrm{diag}\left(\hat{\Sigma}_k\right)^{-1}, \tag{9}$$

corresponding to uniform averaging and Gaussian averaging, respectively, where $\hat{\Sigma}_k$ denotes the standard empirical estimate of the covariance. These are our baselines for comparison.

**Evaluation metrics.** Since the goal of MCMC is usually to estimate event probabilities and function expectations, we evaluate algorithm accuracy for such estimates, relative to serial MCMC output. For each model, we consider a suite of test functions $f \in \mathcal{F}$ (e.g. low degree polynomials, cluster comembership indicators), and we assess the error of each algorithm $\mathcal{A}$ using the metric

$$\epsilon_{\mathcal{A}}\left(f\right) = \frac{|\mathbb{E}_{\mathcal{A}}\left[f\right] - \mathbb{E}_{\mathrm{MCMC}}\left[f\right]|}{|\mathbb{E}_{\mathrm{MCMC}}\left[f\right]|}.$$

In the body of the paper, we report median values of $\epsilon_{\mathcal{A}}$, computed within each test function class. The supplement expands on this further, showing quartiles for the differences in $\epsilon_{\mathrm{VCMC}}$ and $\epsilon_{\mathrm{CMC}}$.

**Bayesian probit regression.** We consider the nonconjugate probit regression model. In this case, we use linear aggregation functions as our function class. For computational efficiency, we also limit ourselves to diagonal $W_k$. We use Gibbs sampling on the following augmented model:

$$\beta \sim \mathcal{N}(0, \, \sigma^2 I_d), \qquad Z_n \mid \beta, \, x_n \sim \mathcal{N}(\beta^T x_n, \, 1), \qquad Y_n \mid Z_n, \, \beta, \, x_n = \begin{cases} 1 \text{ if } Z_n > 0, \\ 0 \text{ otherwise.} \end{cases}$$

This augmentation allows us to implement an efficient and rapidly mixing Gibbs sampler, where

$$\beta \mid x_{1:N} = \mathrm{X}, \qquad z_{1:N} = z \sim \mathcal{N}\left(\Sigma \mathrm{X}^T z, \, \Sigma\right), \qquad \Sigma = \left(\sigma^{-2} I_d + \mathrm{X}^T \mathrm{X}\right)^{-1}.$$

We run two experiments: the first using a data generating distribution from Scott et al. [22], with $N = 8500$ data points and $d = 5$ dimensions, and the second using $N = 10^5$ data points and $d = 300$ dimensions. As shown in Figure 1 and, in the supplement,[1] Figures 4 and 5, VCMC decreases the error of moment estimation compared to the baselines, with substantial gains starting at $K = 25$ partitions (and increasing with $K$). We also run the high-dimensional experiment using partial posteriors [23] in place of subposteriors, and observe substantially lower errors in this case.

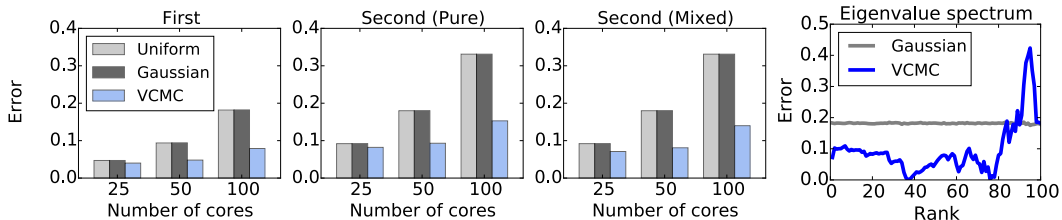

Figure 2: High-dimensional normal-inverse Wishart model ($d = 100$). *(Far left, left, right)* Moment approximation error for the uniform and Gaussian averaging baselines and VCMC, relative to serial MCMC. Letting $\rho_j$ denote the $j^{\text{th}}$ largest eigenvalue of $\Lambda^{-1}$, we assessed three groups of functions: first moments, with $f(\Lambda) = \rho_j$ for $1 \leq j \leq d$; pure second moments, with $f(\Lambda) = \rho_j^2$ for $1 \leq j \leq d$; and mixed second moments, with $f(\Lambda) = \rho_i \rho_j$ for $1 \leq i < j \leq d$. *(Far right)* Graph of error in estimating $\mathbb{E}[\rho_j]$ as a function of $j$ (where $\rho_1 \geq \rho_2 \geq \cdots \geq \rho_d$).

**Normal-inverse Wishart model.**    To compare directly to prior work [22], we consider the normal-inverse Wishart model

$$\Lambda \sim \text{Wishart}(\nu, V), \qquad X_n \mid \mu, \Lambda \sim \mathcal{N}(\mu, \Lambda^{-1}).$$

Here, we use spectral aggregation rules as our function class, restricting to diagonal $W_k$ for computational efficiency. We run two sets of experiments: one using the covariance matrix from Scott et al. [22], with $N = 5000$ data points and $d = 5$ dimensions, and one using a higher-dimensional covariance matrix designed to have a small spectral gap and a range of eigenvalues, with $N = 10^5$ data points and $d = 100$ dimensions. In both cases, we use a form of projected SGD, using 40 samples per iteration to estimate the variational gradients and running 25 iterations of optimization. We note that because the mean $\mu$ is treated as a point-estimated parameter, one could sample $\Lambda$ exactly using normal-inverse Wishart conjugacy [10]. As Figure 2 shows,[2] VCMC improves both first and second posterior moment estimation as compared to the baselines. Here, the greatest gains from VCMC appear at large numbers of partitions ($K = 50, 100$). We also note that uniform and Gaussian averaging perform similarly because the variances do not differ much across partitions.

**Mixture of Gaussians.**    A substantial portion of Bayesian inference focuses on latent variable models and, in particular, mixture models. We therefore evaluate VCMC on a mixture of Gaussians,

$$\theta_{1:L} \sim \mathcal{N}(0, \tau^2 I_d), \qquad Z_n \sim \text{Cat}(\pi), \qquad X_n \mid Z_n = z \sim \mathcal{N}(\theta_z, \sigma^2 I_d),$$

where the mixture weights $\pi$ and the prior and likelihood variances $\tau^2$ and $\sigma^2$ are assumed known. We use the combinatorial aggregation functions defined in Section 5; we set $L = 8$, $\tau = 2$, $\sigma = 1$, and $\pi$ uniform and generate $N = 5 \times 10^4$ data points in $d = 8$ dimensions, using the model from Nishihara et al. [19]. The resulting inference problem is therefore $L \times d = 64$-dimensional. All samples were drawn using the PyStan implementation of Hamiltonian Monte Carlo (HMC).

As Figure 3a shows, VCMC drastically improves moment estimation compared to the baseline Gaussian averaging (9). To assess how VCMC influences estimates in cluster membership probabilities, we generated 100 new test points from the model and analyzed cluster comembership probabilities for all pairs in the test set. Concretely, for each $x_i$ and $x_j$ in the test data, we estimated $\mathbb{P}[x_i$ and $x_j$ belong to the same cluster]. Figure 3a shows the resulting boost in accuracy: when $\sigma = 1$, VCMC delivers estimates close to those of serial MCMC, across all numbers of partitions; the errors are larger for $\sigma = 2$. Unlike previous models, uniform averaging here outperforms Gaussian averaging, and indeed is competitive with VCMC.

**Assessing computational efficiency.**    The efficiency of VCMC depends on that of the optimization step, which depends on factors including the step size schedule, number of samples used per iteration to estimate gradients, and size of data minibatches used per iteration. Extensively assessing the influence of all these factors is beyond the scope of this paper, and is an active area of research both in general and specifically in the context of variational inference [13, 17, 21]. Here, we provide

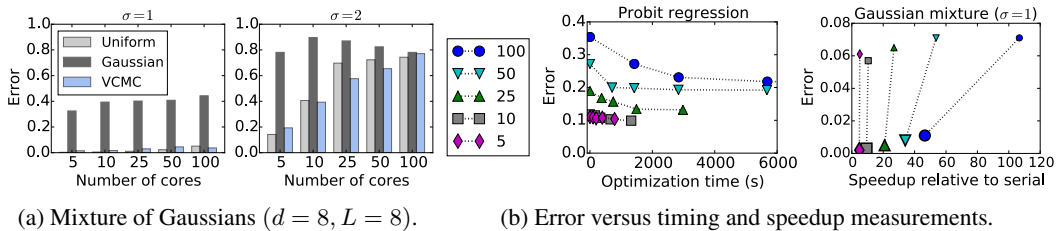

(a) Mixture of Gaussians ($d = 8, L = 8$).　　(b) Error versus timing and speedup measurements.

Figure 3: (a) Expectation approximation error for the uniform and Gaussian baselines and VCMC. We report the median error, relative to serial MCMC, for cluster comembership probabilities of pairs of test data points, for *(left)* $\sigma = 1$ and *(right)* $\sigma = 2$, where we run the VCMC optimization procedure for 50 and 200 iterations, respectively. When $\sigma = 2$, some comembership probabilities are estimated poorly by all methods; we therefore only use the 70% of comembership probabilities with the smallest errors across all the methods. (b) *(Left)* VCMC error as a function of number of seconds of optimization. The cost of optimization is nonnegligible, but still moderate compared to serial MCMC—particularly since our optimization scheme only needs small batches of samples and can therefore operate concurrently with the sampler. *(Right)* Error versus speedup relative to serial MCMC, for both CMC with Gaussian averaging (small markers) and VCMC (large markers).

an initial assessment of the computational efficiency of VCMC, taking the probit regression and Gaussian mixture models as our examples, using step sizes and sample numbers from above, and eschewing minibatching on data points.

Figure 3b shows timing results for both models. For the probit regression, while the optimization cost is not negligible, it is significantly smaller than that of serial sampling, which takes over 6000 seconds to produce 1000 effective samples.[3] Across most numbers of partitions, approximately 25 iterations—corresponding to less than 1500 seconds of wall clock time—suffices to give errors close to those at convergence. For the mixture, on the other hand, the computational cost of optimization is minimal compared to serial sampling. We can see this in the overall speedup of VCMC relative to serial MCMC: for sampling and optimization combined, low numbers of partitions ($K \leq 25$) achieve speedups close to the ideal value of $K$, and large numbers ($K = 50, 100$) still achieve good speedups of about $K/2$. The cost of the VCMC optimization step is thus moderate—and, when the MCMC step is expensive, small enough to preserve the linear speedup of embarrassingly parallel sampling. Moreover, since the serial bottleneck is an optimization, we are optimistic that performance, both in terms of number of iterations and wall clock time, can be significantly increased by using techniques like data minibatching [9], adaptive step sizes [21], or asynchronous updates [20].

## 7　Conclusion and future work

The flexibility of variational consensus Monte Carlo (VCMC) opens several avenues for further research. Following previous work on data-parallel MCMC, we used the subposterior factorization. Our variational framework can accomodate more general factorizations that might be more statistically or computationally efficient – e.g. the factorization used by Broderick et al. [4]. We also introduced structured sample aggregation, and analyzed some concrete instantiations. Complex latent variable models would require more sophisticated aggregation functions – e.g. ones that account for symmetries in the model [5] or lift the parameter to a higher dimensional space before aggregating. Finally, recall that our algorithm – again following previous work – aggregates in a sample-by-sample manner, cf. (4). Other aggregation paradigms may be useful in building approximations to multimodal posteriors or in boosting the statistical efficiency of the overall sampler.

**Acknowledgments.**　We thank R.P. Adams, N. Altieri, T. Broderick, R. Giordano, M.J. Johnson, and S.L. Scott for helpful discussions. E.A. is supported by the Miller Institute for Basic Research in Science, University of California, Berkeley. M.R. is supported by a Hertz Foundation Fellowship, generously endowed by Google, and an NSF Graduate Research Fellowship. Support for this project was provided by Amazon and by ONR under the MURI program (N00014-11-1-0688).

## Footnotes

[1]Due to space constraints, we relegate results for $d = 5$ to the supplement.

[2]Due to space constraints, we compare to the $d = 5$ experiment of Scott et al. [22] in the supplement.

[3]We ran the sampler for $5100$ iterations, including $100$ burnin steps, and kept every fifth sample.

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
