[Supplementary Material · paper-supp.pdf]

# A Proofs

*Proof of entropy relaxation.* We apply the entropy power inequality [6], which asserts that for independent $d$-dimensional random vectors $\psi_{1:K}$, the sum

$$\psi = \sum_{k=1}^{K} \psi_k$$

satisfies

$$e^{\frac{2h(\psi)}{d}} \geq \sum_{k=1}^{K} e^{\frac{2h(\psi_k)}{d}} \geq \max_{1 \leq k \leq K} e^{\frac{2h(\psi_k)}{d}}, \tag{10}$$

where $h$ denotes differential entropy.

In our case, we have

$$\psi_k = F_k(\theta_k)$$

and

$$\psi = \theta = F(\theta_1, \ldots, \theta_K)$$

Since

$$\mathrm{H}[q] = h(\psi),$$

equation (10) implies

$$\mathrm{H}[q] \geq \max_{1 \leq k \leq K} h(\psi_k) = \max_{1 \leq k \leq K} \left( \mathrm{H}[p_k] + \mathbb{E}_{p_k} \left[ \log \det J(F_k)(\theta_k) \right] \right).$$

Defining

$$\tilde{\mathrm{H}}[q] = \frac{1}{K} \sum_{k=1}^{K} \mathbb{E}_{p_k} \left[ \log \det J(F_k)(\theta_k) \right] + \min_{1 \leq k \leq K} \mathrm{H}[p_k],$$

we immediately see that

$$\mathrm{H}[q] \geq \tilde{\mathrm{H}}[q],$$

as required. $\qquad\square$

*Proof of Theorem 4.1.* We first define

$$\mathcal{L}_0(q) = \mathbb{E}_q \left[ \log p(\theta, X) \mid \theta_{1:K} \right] = \log p(F(\theta_{1:K}), X).$$

Since $\mathcal{L}(q) = \mathbb{E}_{p_{1:K}}[\mathcal{L}_0(q)]$, where the expectation is taken with respect to the subposteriors, which do not vary with $q$, it suffices to show that $\mathcal{L}_0$ is concave in each $F^u$ individually for each fixed $\theta_{1:K}$. Furthermore, since $F(\theta_{1:K})$ is linear in $F$ by the definition of function addition, it actually suffices to show $\ell(\theta) = \log p(F(\theta_{1:K}), X)$ in each $\theta^u$ individually. To see why this holds, first observe that for each $u \in V(G)$, we have

$$\ell(\theta) = \log h^u(\theta^u) + \sum_{u' \in \mathrm{par}(u)} \left( \theta^{u'} \right)^T T^{u' \to u} \left( \theta^{u'} \right) \tag{11}$$

$$+ \sum_{v \in \mathrm{ch}(u)} \left[ (\theta^u)^T T^{u \to v}(\theta^v) - \log A^v \left( \theta^{\mathrm{par}(v)} \right) \right] + c_u, \tag{12}$$

where $c_u$ is a function of $\theta$ that is constant in $\theta^u$. By the log-concavity assumption, the sum of the first two terms of $\ell(\theta)$ in (12) is concave in $\theta^u$. On the other hand, by basic properties of exponential families, each $\log A^v \left( \theta^{\mathrm{par}(v)} \right)$ is convex in $\theta^{\mathrm{par}(v)}$ and hence in $\theta^u$, making its negative concave. Since the remaining terms are linear or constant, $\ell$ is in fact concave in $\theta^u$. The claim follows. $\qquad\square$

*Proof of Theorem 4.2.* Clearly it suffices to show that each $\mathbb{E}_{p_k} \left[ \log \det J(F_k)(\theta_k) \right]$ is concave and for this it suffices to show that for fixed $\theta_k$, $\log \det J(F_k)(\theta_k)$ is concave. This is immediate, however, since the Jacobian is a linear function and $\log \det$ is a concave function. $\qquad\square$

# B Variational objective functions

We derive the variational objectives and gradients for the models we analyze. Throughout, we make the convention that for $A, B \in \mathbb{R}^{d \times d}$,

$$\langle\langle A, B \rangle\rangle = \mathrm{Tr}(AB)$$

denotes the trace inner product.

## B.1 Bayesian probit regression

In this section, we compute the variational objective for the Bayesian probit regression model. For convenience, we define

$$\mu_k = \mathbb{E}_{p_k}\left[\beta_k\right] \text{ and } S_k = \mathbb{E}_{p_k}\left[\beta_k \beta_k^T\right].$$

In this notation, the variational objective takes the simple form

$$\mathcal{L}(W) = -\frac{1}{2\sigma^2} \sum_{k=1}^{K} \left[ \left\langle\!\left\langle S_k,\, W_k^T W_k \right\rangle\!\right\rangle + 2 \sum_{\ell \neq k} \left\langle\!\left\langle \mu_k \mu_\ell^T W_\ell^T,\, W_k \right\rangle\!\right\rangle \right]$$

$$+ \sum_{n=1}^{N} \left[ y_n \cdot \mathbb{E}_q\left[\log \Phi_n\right] + (1 - y_n) \cdot \mathbb{E}_q\left[\log\left(1 - \Phi_n\right)\right] \right]$$

$$+ \frac{1}{K} \sum_{k=1}^{K} \log \det\left(W_k\right)$$

where $\Phi_n = \Phi\left(\sum_k \left\langle\!\left\langle W_k,\, \beta_k x_n^T \right\rangle\!\right\rangle\right)$.

This leads to the gradients

$$\nabla_{W_k}\mathcal{L} = \frac{1}{\sigma^2} \left[ S_k W_k^T + \sum_{\ell \neq k} \left( \mu_k \mu_\ell^T W_\ell^T + W_\ell \mu_\ell \mu_k^T \right) \right]$$

$$+ \sum_{n=1}^{N} \mathbb{E}_q\left[ \left( \frac{\phi_n}{\Phi_n\left(1 - \Phi_n\right)} \cdot \left(y_n - \Phi_n\right) \right) \cdot \beta_k \right] x_n^T$$

$$+ \frac{W_k^{-1}}{K},$$

where we have additionally defined $\phi_n = \phi\left(\sum_{k=1}^{K} \left\langle\!\left\langle W_k,\, \beta_k x_n^T \right\rangle\!\right\rangle\right)$ and

$$\beta = \sum_{k=1}^{K} W_k \beta_k.$$

## B.2 Normal-inverse Wishart model

The variational objective for the normal-inverse Wishart model takes the form

$$\mathcal{L}(W) = \mathbb{E}_q\left[\mathcal{L}_0\left(W,\, \Lambda_{1:K}\right)\right] + \tilde{\mathrm{H}}\left[q\right],$$

where

$$\mathcal{L}_0(W) = -\frac{1}{2} \sum_{k=1}^{K} \left\langle\!\left\langle R_k\left(V^{-1} + X^T X\right) R_k^T,\, W_k D_k \right\rangle\!\right\rangle$$

$$+ \frac{N}{2} \sum_{k=1}^{K} \left\langle\!\left\langle R_k\left(\mu \bar{x}^T + \bar{x}\mu^T\right),\, W_k D_k \right\rangle\!\right\rangle - \frac{N}{2} \sum_{k=1}^{K} \left\langle\!\left\langle \left(R_k \mu\right)\left(R_k \mu\right)^T,\, W_k D_k \right\rangle\!\right\rangle$$

$$+ \frac{\nu + N - d - 1}{2} \cdot \log \det\left( \sum_{k=1}^{K} R_k^T \left[W_k D_k\right] R_k \right),$$

and we have compressed our notation by setting $\mu = \sum_k A_k \mu_k$, $\bar{x} = \frac{1}{N}\sum_n x_n$, $R_k = R\left(\Lambda_k\right)$, and $D_k = D\left(\Lambda_k\right)$. As before, we have

$$\tilde{\mathrm{H}}\left[q\right] = \frac{1}{K} \sum_{k=1}^{K} \log \det\left(W_k\right),$$

where we have suppressed the constant depending on the $p_{1:K}$ since it does not vary with $W_k$.

Recalling that $W_k$ is diagonal, we can obtain the gradients by first computing

$$\nabla_{W_k}\mathcal{L}_0(W) = D_k \cdot \mathrm{diag}\left[ R_k\left(V^{-1} + X^T X\right) R_k^T \right]$$

$$+ \frac{N}{2} \cdot D_k\left(R_k \mu \circ \bar{x} + R_k \bar{x} \circ \mu\right) - \frac{N}{2} \cdot D_k\left(R_k \mu\right) \circ \left(R_k \mu\right)$$

$$+ \frac{\nu + N - d - 1}{2} \cdot D_k \cdot \mathrm{diag}\left[ R_k \left( \sum_{\ell=1}^{K} R_\ell^T \left[W_\ell D_\ell\right] R_\ell \right)^{-1} R_k^T \right],$$

where we have used $\circ$ to denote elementwise vector products. We then find

$$\nabla_{W_k}\mathcal{L} = \mathbb{E}_q\left[\nabla_{W_k}\mathcal{L}_0\left(W\right)\right] + \frac{W_k^{-1}}{K}.$$

## B.3  Mixture of Gaussians

Per the description of aggregation in Section 5, we define merged samples in the mixture of Gaussians model by the equations

$$\theta_\ell^* = F_{a\ell}\left(\theta_{1:K,1:L}\right) = \sum_{k=1}^{K} W_{k\ell}\theta_{ka_{k\ell}},$$

where $\ell = 1, \ldots, L$ denotes the cluster index and $a_k$ denotes the alignment mapping indices on the master core to indices on worker core $k$. Throughout this section, we treat the alignment variables as fixed.

Using this notation, we define

$$\mathcal{L}_0\left(W,\ \theta_{1:K,1:L}\right) = -\frac{1}{2\tau^2}\sum_{\ell=1}^{L}||\theta_\ell^*||_2^2 - \frac{1}{2\sigma^2}\sum_{\ell=1}^{L}\sum_{i=1}^{n}\gamma_{i\ell}\left(W\right)||\theta_\ell^* - x_i||_2^2,$$

where

$$\gamma_{n\ell} = \frac{\tilde{\gamma}_{n\ell}}{\sum_{\ell'=1}^{L}\tilde{\gamma}_{n\ell'}}$$

and

$$\tilde{\gamma}_{n\ell} = \exp\left(-\frac{1}{2\sigma^2}||\theta_\ell^* - x_n||_2^2\right).$$

The variational objective then takes the form

$$\mathcal{L}\left(W\right) = \mathbb{E}_{p_{1:K}}\left[\mathcal{L}_0\left(W,\ \theta_{1:K,1:L}\right)\right] + \tilde{H}\left[q\right],$$

with the usual equation

$$\tilde{H}\left[q\right] = \frac{1}{K}\sum_{k=1}^{K}\sum_{\ell=1}^{L}\log\det\left(W_{k\ell}\right).$$

Some calculation then shows that the gradients with respect to the various $W_{k\ell}$ are given by

$$\nabla_{k\ell}\mathcal{L}_0\left(W,\ \theta_{1:K,1:L}\right) = \frac{1}{2\sigma^4}\sum_{n=1}^{N}\gamma_{n\ell}\left(1 - \gamma_{i\ell}\right)||\theta_\ell^* - x_n||_2^2 \cdot \theta_{ka_{k\ell}}\left(\theta_\ell^* - x_n\right)^T$$

$$-\left(\frac{1}{\tau^2} + \frac{\sum_{n=1}^{N}\gamma_{n\ell}}{\sigma^2}\right)\cdot\theta_{ka_{k\ell}}\left(\theta_\ell^* - \tilde{x}_\ell\right)^T,$$

where

$$\tilde{x}_\ell = \left(\frac{1}{\tau^2} + \frac{\sum_{n=1}^{N}\gamma_{n\ell}}{\sigma^2}\right)^{-1}\sum_{n=1}^{N}\frac{\gamma_{n\ell}}{\sigma^2}\cdot x_n.$$

This covers the case of general PSD matrices $W_{k\ell}$. When the matrices are restricted to be diagonal, we get the simplified gradient

$$\nabla_{k\ell}\mathcal{L}_0\left(W,\ \theta_{1:K,1:L}\right) = \frac{1}{2\sigma^4}\sum_{n=1}^{N}\gamma_{i\ell}\left(1 - \gamma_{n\ell}\right)||\theta_\ell^* - x_n||_2^2 \cdot \theta_{ka_{k\ell}}\circ\left(\theta_\ell^* - x_n\right)$$

$$-\left(\frac{1}{\tau^2} + \frac{\sum_{n=1}^{N}\gamma_{n\ell}}{\sigma^2}\right)\cdot\theta_{ka_{k\ell}}\circ\left(\theta_\ell^* - \tilde{x}_\ell\right),$$

where $\circ$ denotes elementwise multiplication of vectors.

Since

$$\nabla_{k\ell}\mathcal{L}\left(W\right) = \mathbb{E}_{p_{1:K}}\left[\nabla_{k\ell}\mathcal{L}\left(W,\ \theta_{1:K,1:L}\right)\right] + \frac{W_{k\ell}^{-1}}{K},$$

this gives us all the information we need to implement an optimization procedure for the objective.

# C   Extended empirical evaluation

Figure 4: Five-dimensional probit regression ($d = 5$). Moment approximation error for the uniform and Gaussian averaging baselines and VCMC, relative to serial MCMC. We assessed three groups of functions: *(left)* first moments, with $f(\beta) = \beta_j$ for $1 \leq j \leq d$; *(center)* pure second moments, with $f(\beta) = \beta_j^2$ for $1 \leq j \leq d$; and *(right)* mixed second moments, with $f(\beta) = \beta_i\beta_j$ for $1 \leq i < j \leq d$.

Figure 5: High-dimensional probit regression ($d = 300$). Moment approximation error for the uniform and Gaussian averaging baselines and VCMC, relative to serial MCMC, for subposteriors *(left)* and partial posteriors *(right)*. Here we show the pure second moments.

Figure 6: Five-dimensional normal-inverse Wishart model ($d = 5$). Moment approximation error for the uniform and Gaussian averaging baselines and VCMC, relative to serial MCMC. Letting $\rho_j$ denote the $j^{\text{th}}$ largest eigenvalue of $\Lambda^{-1}$, we assessed three groups of functions: *(left)* first moments, with $f(\Lambda) = \rho_j$ for $1 \leq j \leq d$; *(center)* pure second moments, with $f(\Lambda) = \rho_j^2$ for $1 \leq j \leq d$; and *(right)* mixed second moments, with $f(\Lambda) = \rho_i \rho_j$ for $1 \leq i < j \leq d$.