[Reviews · NeurIPS 2015]

Submitted by Assigned_Reviewer_1

Consensus Monte Carlo (CMC) is a method for parallelizing MCMC for posterior inference over large datasets. It works by factorizing the posterior distribution into sub-posteriors each of which depend on only a subset of datapoints, sampling from each of these sub-posteriors in parallel, and then transforming samples from the sub-posteriors using an aggregation function to samples from the real posterior. Existing works use very naive methods of aggregation which result in high bias, or are computationally very expensive, which make it difficult to use Consensus Monte Carlo in practice. This paper proposes a more principled way of combining samples by optimizing over aggregation functions using variational inference.

Clarity: The paper is well written and easy to follow.

Significance: Bayesian inference for big datasets is a very important problem. This paper is significant because it presents a way of making parallel MCMC using CMC more practical.

Originality: The novelty in the paper is the use of variational inference to optimize over CMC aggregation functions instead of using a fixed one. This is rather straightforward, except that the entropy term is difficult to estimate and the authors propose minimizing a lower bound of it instead. Nevertheless, this is an important difference and allows the proposed method to achieve a large reduction in error in estimating posterior expectations compared to very simple baselines.

Quality: The method itself seems sound, but I am not quite satisfied with the experiments. All experiments are on toy problems and datasets. Although it is nice to have these toy examples as they illustrate important aspects of the algorithm like the ability to aggregate structured samples, more realistic experiments are also necessary. Parallel MCMC is useful mainly for large datasets, but the authors compare on very small datasets, the largest of which only had 50K datapoints. I also wish the authors had experiments on more interesting models, e.g. LDA rather than on a Gaussian mixture model or normal-wishart, so that their method is of more direct interest to practitioners (plus there are large, real datasets for LDA) Also, the authors compare only against very simple baselines. How does this compare to Neiswanger et al or the Weierstrass sampler method of Wang and Dunson (for settings where these methods are also applicable)? How does this compare to serial mini-batch algorithms like Stochastic Gradient Langevin Dynamics (SGLD)? Although the proposed method could also use SGLD for sampling from sub-posteriors, does the additional time in aggregating samples and optimizing the aggregating function prevent it from having an advantage over vanilla SGLD? Finally, I really wish these methods were applied to a problem where there is a clear advantage of using Bayesian inference as compared to using a point estimate, e.g. where point estimates overfit, or the uncertainty obtained from Bayesian inference is actually used for something.

Summary: This is a good paper and is well written, but the experimental section could be a lot better. I lean towards accepting it, but rejecting the paper to wait for more convincing and realistic experiments would not be a big loss.

Submitted by Assigned_Reviewer_2

I found the methodology and experiments presented in this paper fairly convincing and have only one some minor comments.

If (as is the case for many applications involving large datasets) there is a focus on predictive inference it may make more sense to aggregate functions of the parameters of relevance for prediction rather than the full set of parameters.

For example, in the probit regression one could aggregate fitted probabilities and in the mixture of Gaussians example one could do something similar for cluster membership probabilities.

I wonder whether the results are more or less sensitive to the aggregation method in such a case.

Have the authors done any experiments along these lines?

Another question that occurred to me was about the structured aggregation for the case of positive semidefinite matrices;

there are certainly different reparametrizations that could be used here (such as a Cholesky factorization) and I wondered why the aggregation is only being done on the D(\Lambda_k) matrices.

Is there any reason why this would be the best choice?

Minor comment:

there is a missing integration in equation (3).

Summary: This paper refines recently suggested consensus Monte Carlo algorithms suggested in the literature by using variational Bayes methods within the aggregation step of these algorithms.

The relaxation of the variational objective function suggested is clever and I found the experiments convincing that the proposed approach produces some improvements over simpler methods.

Submitted by Assigned_Reviewer_3

Paper Title: Variational Consensus Monte Carlo

Paper Summary: This paper presents a new method for aggregating samples in a low-communication MCMC setting. In this setting, a large dataset is partitioned over multiple cores, MCMC is performed on a subset of data on each core (technically, samples are drawn from a subposterior distribution on each core), and the resulting samples are aggregated with some combination function to produce samples from the full posterior distribution. Instead of developing a fixed combination function, as previous methods have done, this paper generalizes the Consensus Monte Carlo [Scott et al., 2013] combination function to define a family of combination functions, and performs variational inference over this family to find the function that yields samples from the best approximation (within the family) to the full posterior distribution. One advantage of this method is that it may allow for easier combination of more sophisticated mathematical objects (instead of simply vectors) due to the generalized form of the aggregation. Experiments are shown on a Bayesian probit regression model, a normal-inverse Wishart model, and a mixture of Gaussians.

Comments:

- I feel that the goal of developing scalable methods for Bayesian inference that maintain a good approximation to the posterior distribution is important, and that this paper takes a good step towards this end. I also feel that this the paper provides a clever way of viewing an existing (low-communication) parallel MCMC strategy in terms of optimizing over a variational objective (another, typically separate, strategy in approximate Bayesian inference).

- The primary goal of these low-communication parallel MCMC methods is to reduce the time of inference (while maintaining a good approximation of the posterior distribution), particularly relative to communication-based parallel MCMC methods. This paper does a good job of presenting a method that provides a good approximation of the posterior distribution (compared with existing methods), though there are relatively few experiments verifying that this method maintains the increased speed benefits of low-communication MCMC (relative to existing low-communication parallel MCMC methods, and communication-based parallel MCMC methods). This paper does indeed show comparison against speedups attained by the CMC method in one model, though a more thorough investigation would be nicer.

- I wonder how the presented VCMC method would compare (in terms of inference speed and error) against serial methods for variational inference. VCMC was presented as a "variational Bayes algorithm", though all comparisons were against MCMC methods, and not variational Bayes methods. At its heart, this method is optimizing an approximate posterior (formed via distributed sampling), so it seems natural to compare against variational inference methods. This is particularly important given the recent popularity of scalable variational inference methods (such as stochastic gradient variational inference). Furthermore, there have recently been a few presented methods for low-communication parallel variational inference, which would make good experimental comparisons---see the following two references: (1) Broderick, Tamara, et al. "Streaming variational bayes." Advances in Neural Information Processing Systems. 2013. (2) Campbell, Trevor, and Jonathan P. How. "Approximate Decentralized Bayesian Inference." UAI. 2014.

- One downside of this method is that it has little ability to make any guarantees about the correctness of the final aggregated samples (which some of the current scalable MCMC methods attempt to do); it simply generates samples using the the best combination function chosen out of a pre-specified family of functions. On the other hand, I suppose this isn't much worse than most variational inference methods (which simply choose the best posterior approximation from a family of distributions). Furthermore, this method should, in general, produce better results than CMC method [Scott et al., 2013] (assuming the weighted-average combination function is in the pre-specified family of functions).

- A few theoretical results were shown in this paper (blockwise concavity under certain conditions, and consequences of this). It would be nice to include more discussion regarding the purpose of these results, and why it is beneficial to prove them (perhaps also mentioning what types of concavity results are typically shown in variational inference literature); currently, they are added in the paper without much comment as to why.
Summary: I feel that this paper makes good steps towards the goal of developing scalable approximate Bayesian inference methods (specifically, low-communication parallel methods) that maintain a good posterior approximation. Additionally, optimizing over the sample aggregation function is a clever idea, and seems to produce good results (relative to CMC). However, a more-thorough empirical exploration into inference times/speedups, and comparisons with other methods (particularly variational methods), would strengthen this paper.

Author Feedback
Author rebuttal: We thank the reviewers for their helpful comments. We organize our response by topic.

Originality: In addition to introducing variational optimization to the CMC framework (noted by R1), we also introduce structured aggregation functions that are necessary for executing CMC in the context of constrained parameters (see Section 5).

Experimental Results:

We agree with the reviewers that a number of interesting directions for the experimental evaluation remain open.

(1) Choice of datasets and models.

In choosing the evaluation datasets, we set two goals: (1) direct comparison to previously published consensus Monte Carlo results and (2) analyzing parallel MCMC for higher dimensional problems.

A key issue, we believe, in choosing benchmark problems is the interaction between dimensionality and dataset size. We avoided large data set sizes with small dimensions in order to stay out of the asymptotic Bayesian central limit theorem regime. Truly large scale experiments involving both large data set size and high dimension would be an excellent topic for future work.

As for more complex models -- as we discussed in our conclusion section, dealing with complex latent variable models like LDA is also an open problem, due to the need for more sophisticated aggregation functions. Our approach to the mixture model takes a step in that direction. We will add a remark addressing this problem and its possible resolutions -- e.g. based on the recent work of Campbell et al.

(2) Comparisons to other published work.

In order to isolate the benefit of the variational optimization, we chose to focus on a thorough comparison to the consensus Monte Carlo baseline. Further evaluation against other parallel algorithms would, we believe, be a good addition to an expanded version of our paper.

Likewise, comparisons to minibatch approaches like SGLD, or purely variational Bayes algorithms like SDA or Campbell et al, would be good additions to future work. We view these as being beyond the scope of our current investigation, however. As we understand it, it is still an open problem to determine where each of these competing methods is advantageous.

(3) Evaluating speedup.

Our empirical results demonstrate a meaningful speedup over serial MCMC. In the case of the Gaussian mixture model, we demonstrated a linear speedup. In the case of the probit model, the speedup was sublinear but, depending on the error tolerated, still substantial.

The speedup relative to serial MCMC depends on several factors, including the sampling algorithm, the optimization algorithm, and the parameters of both (e.g. proposal kernels, step size, number of samples, burn in). Further investigation of this high dimensional space is warranted.

(4) Predictive inference evaluation

As suggested by R2, predictive ability is a good evaluation strategy. We deployed it for the mixture of Gaussians, where we showed greatly increased accuracy in estimating cluster comembership probabilities (Figure 3a).

(5) Error metric.

As R5 notes in the brief comment, other error metrics could be used. It's unclear which is best, and we believe ours is a reasonable choice. It has several desirable properties, including, most importantly, scale-freeness.

Theoretical Results:

(1) Blockwise concavity

We emphasized the blockwise concavity of the variational optimization problem as evidence of its tractability.

(2) Alternate relaxations

There are two challenges related to the relaxation: first, finding any relaxation (as the setup is different from classical variational Bayes problems) and, second, assessing the quality of the relaxation and its impact on performance (as R6 notes).

R6 makes a good point that the relaxation is an important choice within the algorithm. Finding any valid relaxation is difficult because the setup is different from classical variational Bayes. We will
add a remark about this in our paper.

Diagonal Matrix Aggregation:

To address R2's question about alternate decompositions, the choice of diagonal-based aggregation was motivated primarily by computational tractability. Alternate aggregation strategies based on, e.g., Cholesky decompositions, as suggested, would certainly be possible.